# Impact Toughness of Spring Steel after Bainite and Martensite Transformation

**Min-Soo Suh** [1], **Seung-Hoon Nahm** [2], **Chang-Min Suh** [3],*  **and No-Keun Park** [4]

1   Korea Institute of Energy Research, Daejeon 34129, Republic of Korea; mssuh@kier.re.kr
2   Korea Research Institute of Standards and Science, Daejeon 34113, Republic of Korea; shnahm@kriss.re.kr
3   School of Mechanical Engineering, Kyungpook Nat'l University, Daegu 41566, Republic of Korea
4   School of Materials Science and Engineering, Yeungnam University, Gyeongsan 38541, Republic of Korea; nokeun_park@yu.ac.kr
*   Correspondence: cmsuh@knu.ac.kr

**Abstract:** It has been reported that a multiphase microstructure with bainite, martensite, and retained austenite obtained by austempering, or quenching and tempering of spring steel containing Si, Mn, and Cr exhibits high strength and ductility. However, little research has been conducted on the bainite formation and impact fracture behavior of next-generation spring steel from the perspective of engineering and industrial applications. The microstructural transformation characteristics of bainite and martensite related to the heat treatment cycle on the maker side were quantitatively analyzed using electron backscatter diffraction (EBSD) and scanning electron microscopy (SEM) analyses. Moreover, the effects and mechanical properties of bainite and martensite formation in response to changes in lath length and width were studied and analyzed. That is, to obtain the mechanical properties of spring steel with the highest quality, tensile and impact specimens, whose microstructure and notch shape change according to the heat-treatment cycle, were prepared and studied.

**Keywords:** spring steel; electron backscatter diffraction; bainite; martensite; austempering; lath; impact test





## 1. Introduction

Recent studies have reported that the multiphase microstructures of bainite, martensite, and retained austenite show high strength and ductility through related studies on the double microstructure and mechanical properties of quench and tempered (QT) and austempered specimens of spring steels [1–12].

The final microstructure of the spring steel produced using the QT method has a tempered martensite structure; however, the problem of hydrogen embrittlement and cracking inevitably occurs when the strength is increased [5–7]. In the case of spring steel, when a delayed hydrogen fracture occurs, the fatigue strength sharply decreases owing to grain boundary brittleness. Therefore, it is expected that the mechanical properties can be improved when the final microstructure of the spring steel is replaced with a mixed structure with bainite rather than martensite [13–16].

Consequently, there is a need for a heat treatment technique that can be applied to products of ultra-high-strength and ultra-long life of grades of Giga steel or higher, which has become a key issue in recent years [1,4,14–16]. However, very few studies have quantitatively analyzed the characteristics of impact tests using microstructures in response to changes in the lath length and width of bainite and martensite by a heat treatment cycle using spring steel [13–16].

In addition, the microstructure had a significant effect on the mechanical properties. Among the various methods of observing microstructures, electron backscattered diffraction (EBSD) has received significant attention. When analyzing the crystal orientation, a much larger area can be captured than when using a transmission electron microscope,

which has been widely used in the past, resulting in statistically good results. That is, it is conveniently used to trace and reproduce the prior austenite grain boundary using EBSD, as well as to reveal the characteristics of the crystal orientation relationship [8–12].

Therefore, in this study, the characteristics of the microstructural transformation related to the heat-treatment cycles were quantitatively analyzed, and the impact and mechanical strength characteristics were studied from the viewpoint of engineering and industrial applications. That is, to obtain the material properties of the spring steel, tensile specimens and two types of impact specimen, whose microstructure and notch shape were changed according to the heat treatment cycle, were produced. The impact fracture behavior and mechanical properties of martensite and bainite microstructures were studied from the perspective of engineering and industrial applications through EBSD and scanning electron microscopy (SEM) analysis.

The QT martensite specimen (quench and tempered) and the austempered bainite AT specimens (subjected to austempering) were prepared by changing the microstructure of SAE9254 (similar to SUP12) AR material (the specimen before heat treatment) according to the heat-treatment cycles.

In the highlights of this study, the results of recent attempts by the manufacturer to improve the performance and quality of spring steel through a microstructural transformation using a new heat-treatment cycle were quantitatively analyzed from the viewpoint of fracture mechanics. Herein, we analyzed the process of transforming bainite to improve its spring life and quality and systematically studied the influencing characteristics and mechanical property changes according to the lath length and the width of the microstructure.

Consequently, the following studies were conducted: (1) the change in impact toughness and mechanical strength characteristics according to the bainite and martensite formation of spring steel, which is important for engineering and industrial applications but on which there is little published data; (2) quantitative analysis of lath length and width after bainite and martensite formation; (3) quantitative analysis of phase transformation properties by EBSD and SEM; (4) comparative analysis of impact toughness characteristics according to the notch shape by fracture surface analysis; and, (5) comparative analysis of bainite and martensite formation with a pole figure.

## 2. Material and Experimental Methods

### 2.1. Test Material and Method of Heat Treatment

The spring steel used in this study was SAE9254, and the chemical composition of the AR spring steel was 0.55 C, 1.5 Si, 0.7 Mn, and 0.7 Cr. QT martensite specimens and AT bainite specimens were obtained by salt bathing following the heat treatment cycle, as shown in Figure 1.

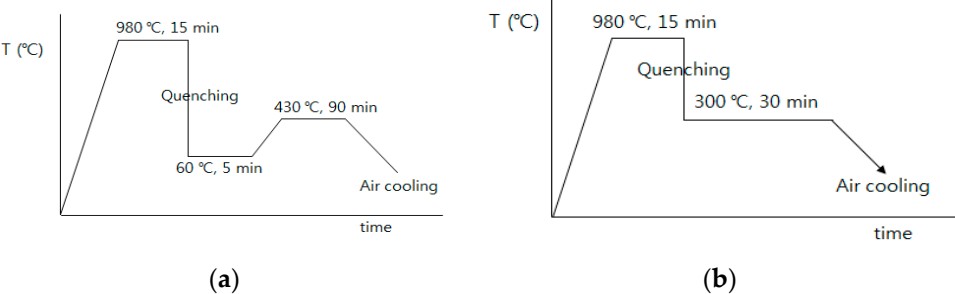

**Figure 1.** Heat treatment cycles used in this study. (**a**) quench and tempered (QT) cycle and (**b**) austempered (AT) cycle.

Round AR spring steel was cut to a diameter of 13 mm and a length of 110 mm. The QT heat treatment used in this study was austenitized in the air at 980 °C for 15 min, water-quenched at 60 °C for 5 min, salt bathed at 430 °C for 90 min, and then air-cooled, as shown in Figure 1a. Subsequently, the same specimen was austenitized at 980 °C for 15 min, salt bathed at 300 °C for 30 min, and then air-cooled, as shown in Figure 1b for austenitization.

Since the Ms temperature of the specimen was evaluated as $265 \pm 4$ °C and the AC line temperature as $755 \pm 5$ °C, martensite transformation did not occur during austempering. These heat-treatment cycles were suggested by the manufacturer. Tensile and impact specimens were prepared using the heat-treated specimens by precision machining.

### 2.2. Experimental Methods

The spring steel was machined into a tensile test specimen (ASTM A370). Tensile tests were performed using a universal testing machine (Autograph AGS-X, Shimadzu, Japan) at a strain rate of 0.02 mm/s according to the specifications of ASTM A370.

The hardness of the specimen was measured using a Vickers hardness tester (JP/HM-112, Mitutoyo, Kawasaki, Japan) and measured 12 times under a 5 kN load, primary and secondary, based on the center. In addition, to observe the microstructure and measure the hardness, we selected a part of the test specimen that was not stressed or deformed. An etching solution of 5 mL HCl, 1 g picric acid, and 100 mL ethanol (95%) was used to observe the optical microstructure (OM).

EBSD of the sample was performed after electropolishing with a mixed solution of 10% perchloric acid and 90% acetic acid. No additional cleaning processes were required. In this study, an EBSD measurement system (DigiView EBSD Camera, EDAX, Mahwah, NJ, USA) was used, and measurement conditions with an accelerating voltage of 18 kV were used. The crystal size was set to 0.3 μm or larger. The fractured impact specimens were magnified and analyzed by fracture surface analysis using an SEM (S-4200, Hitachi, Japan).

Figure 2 shows the notch shapes and dimensions of the impact specimens. Figure 2a shows the shape of a standard V-type notch impact specimen (Korean Standards (KS) specification) with a notch depth of 2 mm and notch radius R of 0.25 mm. The overall size of the impact specimen was L 55 × W 10 × H 10 mm. The sharp I-type notch impact specimen proposed by the author [17] was designed and manufactured as a standard impact specimen with the same dimensions and a notch radius R of 0.125 mm, which was twice as sharp as the standard specimen.

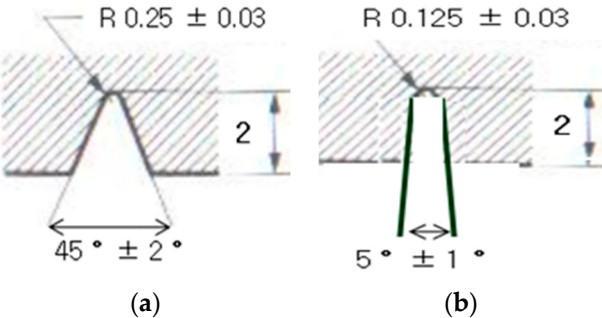

**Figure 2.** (**a**) Notch shape of the standard V-type impact specimen (Korean Standards (KS) specification); (**b**) sharp I-type notch impact specimen (unit: mm) proposed by the author [17].

The notch shape of the impact specimen was precisely machined using a wire cutter. The experiment of the sharp I-type notch impact specimen was aimed to create a more sensitive notch than the standard V-type notch impact specimen. Fatigue cracks usually have a very sensitive crack tip with R < 0.01 mm.

The impact test was performed using a pendulum-type Charpy impact tester (MPX-700, Instron, Norwood, MA, USA) incorporated with an instrumentation sensor. The correlation between the impact toughness characteristics according to microstructural changes caused by heat treatment and the change in impact toughness owing to the notch shapes of the AR, QT, and AT specimens were quantitatively analyzed and examined.

Unlike conventional impact testers, the impact tester used in this study with instrumentation sensors collected quantitative data in a short time (ms) during impact testing and accurately analyzed the impact energy (J) and impact time (ms). It is possible to collect

an accurate analysis of impact conditions and accurate analytical and digital data in the early stages of impact.

## 3. Results and Discussions

### 3.1. Variation in Microstructure Using the Heat Treatment Cycles

Figure 3a shows an optical microscope (OM) photograph of the AR specimen, showing a mixed structure of ferrite (white color), pearlite (gray color), and cementite with an average grain diameter of approximately 11 μm. Figure 3b shows the OM structure of the QT specimen, which shows a refined microstructure after the heat treatment cycle of Figure 1a. Some ferrites were formed owing to compositional heterogeneity, and the rest were transformed into martensite, resulting in white areas appearing in the QT specimen.

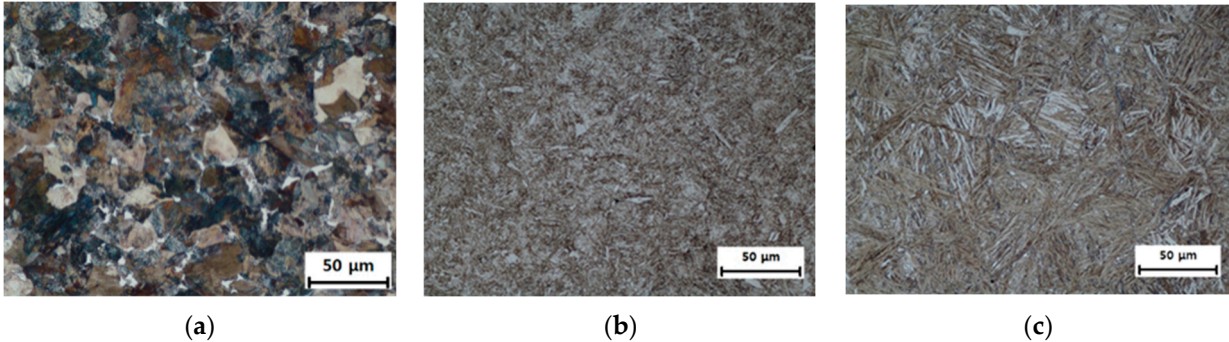

**(a)**      **(b)**      **(c)**

**Figure 3.** Examples of optical microscopy (OM) photographs of three specimens. (**a**) AR (the specimen before heat treatment), (**b**) QT, and (**c**) AT.

Figure 3c shows the OM structure of the AT specimen after the heat treatment cycle shown in Figure 1b. In the AT specimen, the laths (see Figure 7) developed well within a prior austenite grain boundary (PAGB) owing to the microstructural formation, with bainite predominantly formed.

### 3.2. Quantitative Microstructure Analysis Using Electron Backscatter Diffraction (EBSD)

3.2.1. AR Specimen

Figure 4 shows the AR specimen data using EBSD, and the average grain size of the microstructure is approximately 11 μm. Figure 4a shows an inverse pole figure (IPF), and Figure 4b shows a phase map showing the crystal orientation with isotropic grains, and two phases of 76.9% ferrite (red color) and 23% $Fe_3C$ (green color). These figures show that the microstructures of the AR specimen of the spring steel used in this study have a uniform orientation.

3.2.2. Martensite Structure of Quench and Tempered (QT) Specimen

Figure 5 shows the image quality (IQ) picture and IPF as the EBSD results of the QT specimens. The part shown in black does not satisfy the condition that the particle size is at least 0.3 μm during the filtering process.

As shown in Figure 5a, plate-like structures with lengths of 2 to 10 μm were formed owing to the inhomogeneity of the local chemical composition. However, most structures were lath-shaped martensite with a width of less than approximately 1 μm. At the time of EBSD measurement, the body-centered cubic (BCC) structure and face-centered cubic (FCC) structure were used for shooting; however, over 99.6% were BCC. This means that when the specimen was cooled to 60 °C, all existing austenites underwent a phase transformation in martensite.

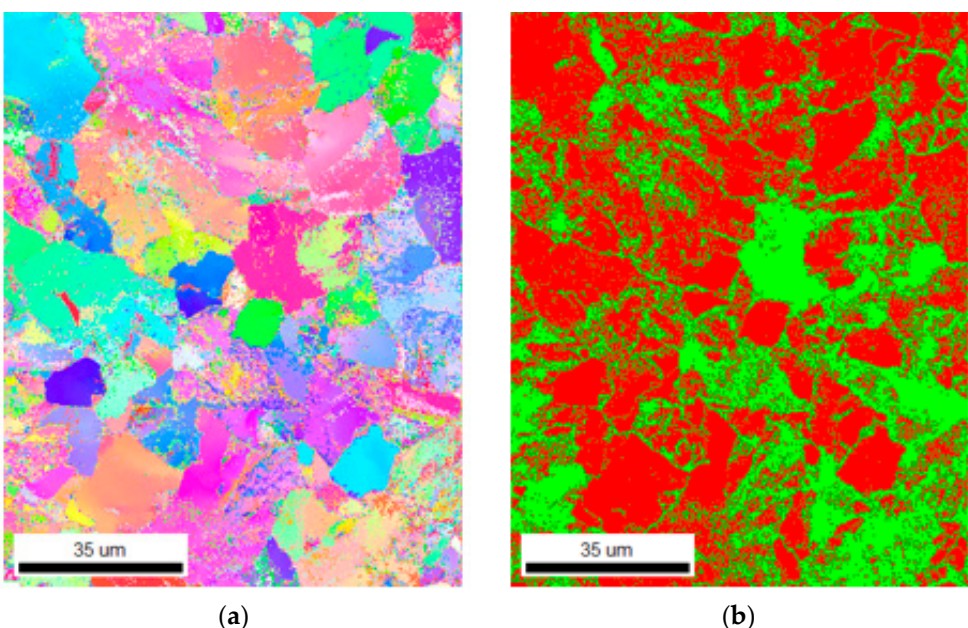

(**a**)        (**b**)

**Figure 4.** Cross-sectional electron backscatter diffraction (EBSD) analysis of the AR specimen. (**a**) inverse pole figure (IPF) map and (**b**) phase map.

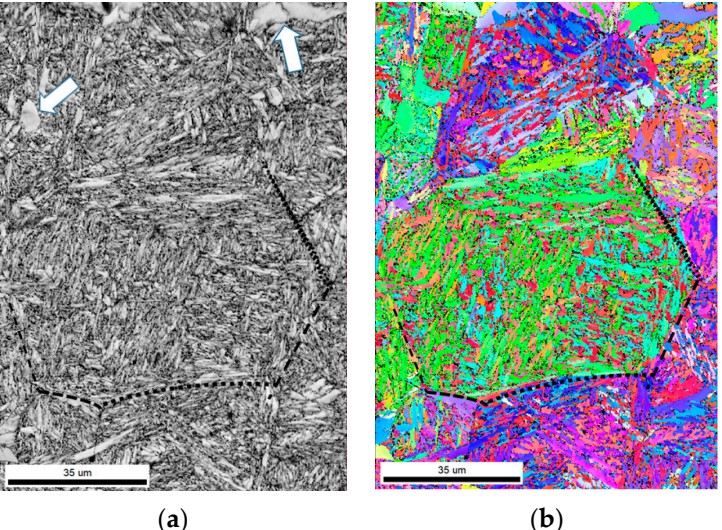

(**a**)        (**b**)

**Figure 5.** EBSD maps of QT specimen: (**a**) image quality (IQ) map and (**b**) IPF map.

The IPF photo in Figure 5b reveals martensite consisting of laths with a maximum number of 24 cases from one PAGB. The dotted line in the photograph represents one PAGB. This made it possible to track the grain boundary, which was the parent phase, immediately before water cooling [11].

Figure 6 is a comparison of the same part as the QT material taken with SEM photograph (a) and EBSD (b). Region A and region B are indicated by arrows in Figure 6, respectively. The white parts in Figure 6a were the cementite that appeared during the tempering process.

In Figure 6b, the red regions are the low-angle boundary [8–12], and the blue regions are the high-angle boundary (15° or more). In region A, many white-looking cementites were observed (Figure 6a). In Figure 6b, many laths of approximately 0.6 μm in width and 2.2 μm in length were formed in region A. Conversely, the density of cementite in region B was low compared to region A in Figure 6.

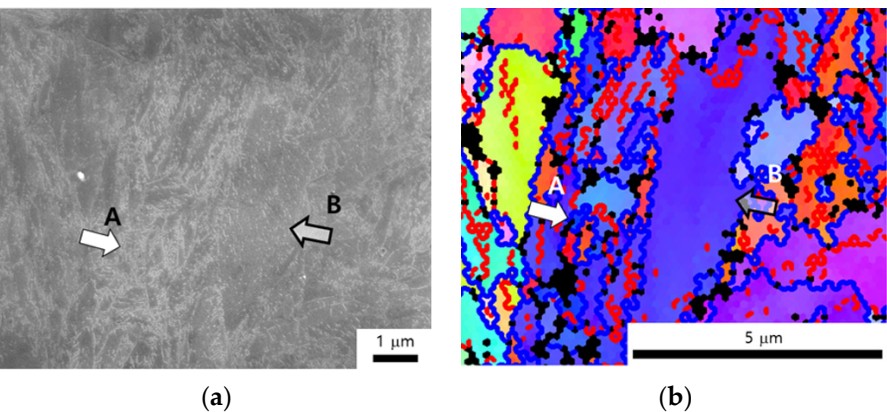

**Figure 6.** SEM image (**a**) and IPF map (**b**) of QT specimen. The white regions in (**a**) represent cementite.

Meanwhile, ferrite grains of approximately 2 μm in width and 4 μm in length appeared (Figure 6b). Although regions A and B were expressed in the same austenite owing to the heterogeneity of the local chemical composition, they had different lath sizes, as shown in Figure 6, and showed a considerable difference in cementite density.

When the initially received specimen was austenitized, phase transformation occurred in the microstructure consisting of proeutectoid ferrite and pearlite to austenite. At this time, the proeutectoid ferrite had a low carbon concentration, whereas the pearlite vicinity had a high carbon concentration. If sufficient carbon diffusion was not performed during the austenite treatment, it had excellent hardenability in A, which was the region where the carbon concentration was relatively high during water cooling. Consequently, many small laths occurred in the prior austenite grain boundaries. It is believed that ferrite phase transformation will occur in region B, where the density of cementite and the amount of carbon are relatively small.

Although coarser than region A, region B also appeared to have an orientation relationship with the prior austenite grain, and it was believed that it was formed at a lower temperature than the temperature exhibited in the diffusion transformation region. The carbon that could not be removed later became cementite in the tempering process, and the local difference in carbon concentration influenced the distribution of cementite.

In region A, a considerable amount of cementite was distributed inside the lath, and it occupied many lath boundaries. In addition, the cementite appeared in the high-angle boundary rather than the low-angle boundary. In the case of high-angle boundaries, it is known that there is a relatively large space in which several atoms can enter between laths [8–12].

It is also known that atomic diffusion occurs very quickly along high-angle boundaries, such as pipe diffusion. Cementite has a high density in the high-angle boundary where diffusion is fast because it can be easily formed with less surface energy during carbon diffusion and cementite formation in the high-angle boundary. In the future, a quantitative study on the distribution of cementite according to the angle between laths is needed.

3.2.3. Bainite Structure of Austempered AT Specimen

(1)　　EBSD Maps of Bainite Structure

Figure 7 shows IQ map (a) and phase map (b) using EBSD made of AT bainite specimen. Some black regions are the parts deleted by filtering. The region indicated by the dotted line represents the region of a PAGB using the crystal orientation relationship [11].

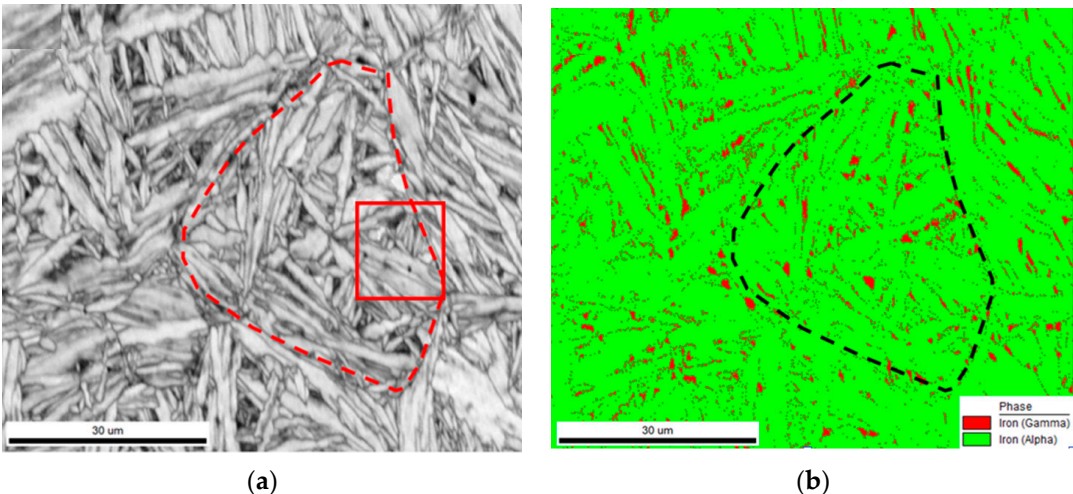

(**a**)           (**b**)

**Figure 7.** EBSD maps of AT specimen: (**a**) IQ map and (**b**) phase map. The dashed line indicates prior austenite grain boundary (PAGB).

Figure 7a shows a plate-like structure with a length of 7 to 29 µm owing to the heterogeneity of the chemical composition. However, most of the microstructure shows a lath-shaped bainite structure with a width of approximately 1.76 µm or less and a length of approximately 15.5 µm. From these data, the length and width of the lath of the AT specimen are longer and wider than those of the QT specimen. The AT material with a large lath length and width is closely related to the improved mechanical properties of the QT material.

In Figure 7b, the phase fraction of FCC is approximately 6.5%, which is much larger than the result, which is less than 1% of that in Figure 5 of the QT specimen. Conversely, the BCC phase fraction appeared to be 93.5%, and the FCC phase (Figure 8a) was approximately 6.5%. Therefore, the bainite AT specimen has more retained austenite than the martensite QT specimen.

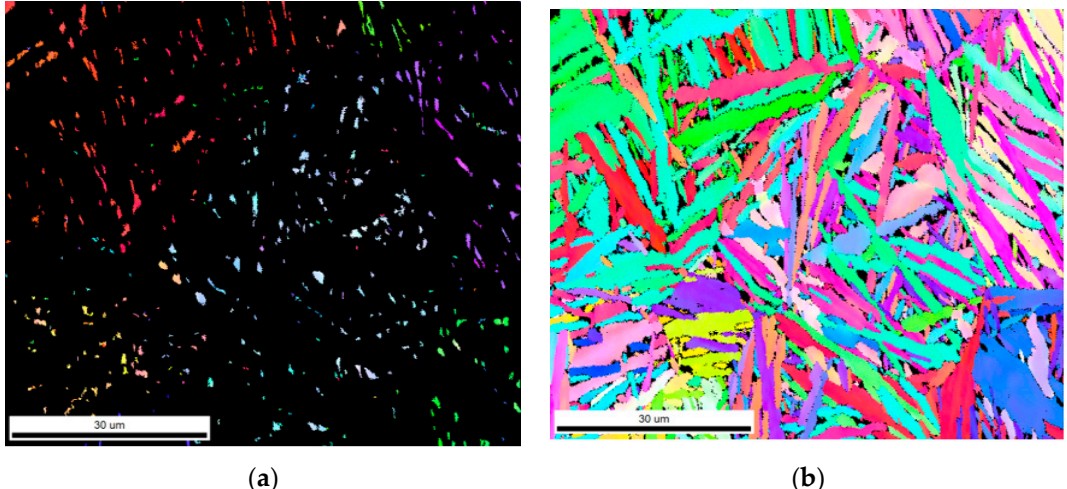

(**a**)           (**b**)

**Figure 8.** EBSD inverse pole figure maps of AT specimen: (**a**) face-centered cubic (FCC) and (**b**) body-centered cubic (BCC).

Figure 8 shows the IPF maps of FCC and BCC of the same part of Figure 7. The former austenite region (inside the dotted line in Figure 7) has a locally similar crystal orientation. Figure 8b is similar to Figure 7b, which illustrates the shape and orientation of bainite made of lath in various cases. However, a phenomenon with a slightly different internal crystal orientation can also be confirmed.

This can consider two degrees of possibilities.

First, different austenite crystals developed during the austenitization process, and a phase transformation occurred during cooling. Different orientations are clearly represented by different colors.

Second, it is the behavior inside the same prior austenite crystal during the phase transformation. That is, when bainite is generated by the phase transformation phenomenon, plastic working is applied to the surrounding austenite, which causes the local crystal orientation of austenite to be transformed. This can be analyzed by observing the orientation of retained austenite along with a method of tracking the PAGB with the Kurdjumov–Sachs orientation relationship.

(2)    Nucleation of Bainite and Retained Austenite

Figure 9 is an enlarged SEM photograph and EBSD results of the rectangular region of Figure 7a, respectively. Moreover, the red dotted line in Figure 9a is the same part as the top right of the PAGB in Figure 7. However, this shows a different characteristic from the martensite QT specimen. That is, a large amount of red-retained austenite appears at the interface of the PAGB as indicated by the ellipse in Figure 9c.

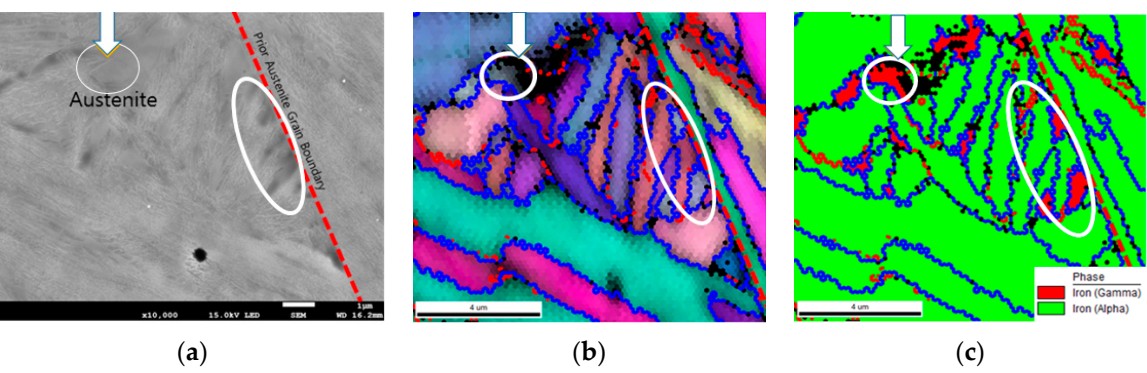

(**a**)                    (**b**)                    (**c**)

**Figure 9.** Scanning electron microscopy (SEM) image (**a**) and EBSD maps (**b**,**c**) of the parts in Figures 7 and 8. (**b**) IPF map combined with an IQ map of the same region with (**a**). (**c**) Phase map of (**b**). The red and green colors indicate the FCC and BCC phases, respectively.

This is also shown in the SEM image in Figure 9a and the EBSD result in Figure 9b, and the parts that are well represented symbolically are marked with a white arrow and a circle. Unlike the QT specimen, a large amount of retained austenite is observed in the bainite AT specimen because of carbon diffusion. That is, when bainite is generated in the phase transformation process, the lath of bainite grows and releases carbon to the outside, and the phase transformation in the vicinity is diffused into austenite that has not yet occurred.

The austenite containing a large amount of carbon, diffused in this method, is further stabilized and remains at a low temperature, such as room temperature. Another microstructural characteristic is shown in Figure 9c, and the retained austenite was found downward, near the PAGB. This phenomenon is illustrated in Figure 9a. This is where the growth of the bainite ended.

Conversely, it can be determined that bainite formation occurred on the right side of the boundary. That is, not all PAGBs are consumed for the nucleation of bainite, and only a specific location can serve as a nucleation site. Meanwhile, a considerable amount of carbon forms cementite inside the lath during bainite growth or tempering.

Figure 9a does not have a locally different distribution of cementite as in Figure 7a; however, it has a homogeneous overall distribution. Cementite does not appear in the retained austenite; however, it is dissolved in austenite and serves to thermodynamically stabilize the austenite. These homogeneous distributions of cementite act as a cause of high strength by providing considerable resistance to the movement of dislocations [18].

(3)    Crystal Orientation Characteristics between The BCC and FCC Phases of Bainite Structure

Figure 10 shows an analysis of the crystal orientation characteristics between the BCC and FCC phases present in one PAGB, as indicated by the dotted line in Figure 7. Figure 10a is an IQ map, where red is the low-angle boundary and blue is the high-angle boundary. As will be described later, Figure 10a is unlike Figure 6b, which shows that it contains a high-angle boundary of a high fraction.

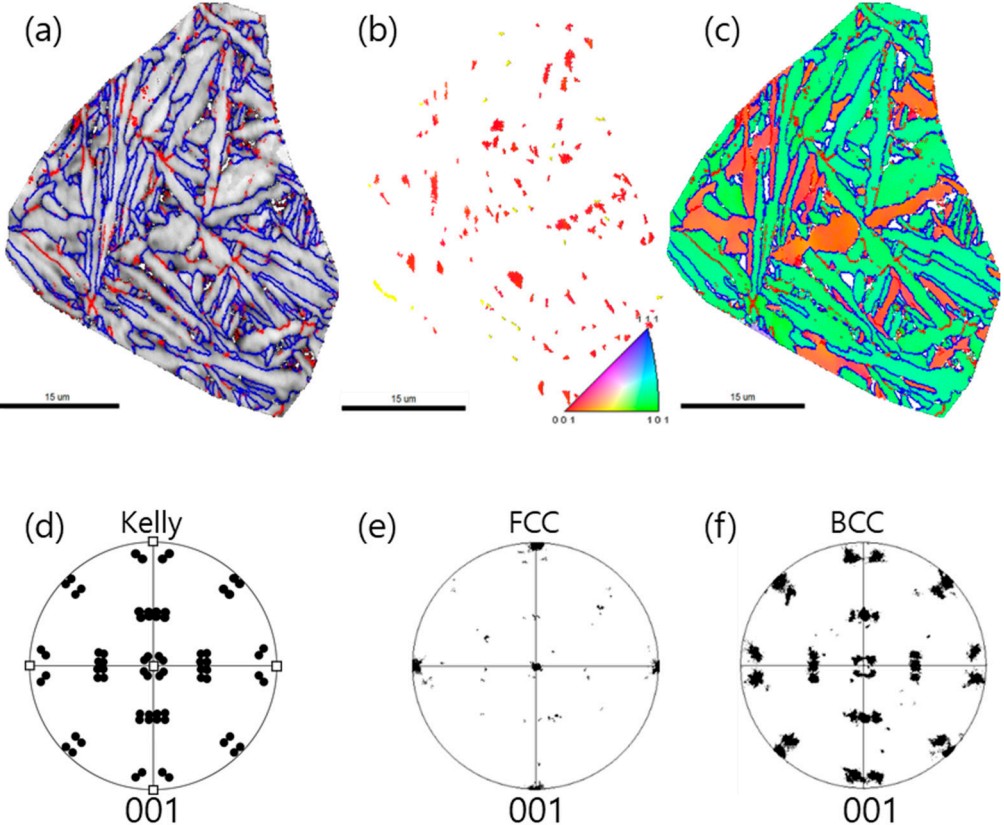

**Figure 10.** (**a**) IQ map of the region selected from Figure 7 (AT specimen). (**b**) IPF map of FCC phase. (**c**) IPF map of BCC phase. (**d**) 001 pole figure of the Kelly orientation relationship between FCC and BCC. The FCC is marked as squares, and the 24 variants of BCC are represented as circles. (**e**) 001 pole figure of (**b**). (**f**) 001 pole figure of (**c**).

Figure 10b shows the same IPF map of the FCC phase, as shown in Figure 10a; however, it is obtained after rotating the crystal in a direction perpendicular to the screen (001). Figure 10c shows the same IPF map of the BCC phase, as shown in Figure 10b, and because it is after the crystal has been rotated, the color expressed differs slightly.

These two phases are known to have Kurdjumov–Sachs, Nishiyama–Wasserman, or Kelly orientation relationships. The theoretical 001 pole figure of the Kelly relation is shown in Figure 10d. That is, the FCC phase is rectangular, and the BCC phase theoretically has 24 types of laths represented in a circle [11–15].

Figure 10e shows the pole figure (001) of the FCC phase shown in Figure 10b and is also distributed as (001). Moreover, some FCCs of other orientations have been observed in Figure 10a. In addition, the pole figure in Figure 10c is shown in Figure 10f. Consequently, in this study it is determined that austenite and martensite have a Kelly orientation relationship, rather than the known Kurdjumov–Sachs orientation relationship.

The Kelly relationship is used because the (111) BCC is distorted by approximately 2.5° in a parallel relationship with the (101) FCC and is closely distributed between the two variants. Thus, the AT specimen is a typical bainite phase transformation with an austenite Kelly relationship during the heat treatment process at 300 °C.

In addition, the difference between the fractions of the low-angle boundary and the high-angle boundary of martensite (Figure 6b) and bainite (Figure 10a) was compared with

the misorientation shown in Figure 11. Both QT and AT specimens have very low distributions of approximately 22° and 45°, respectively. This is because the phases produced from austenite do not appear in the orientation relationship (11). This figure can be used to easily predict PAGB and can also be extracted locally inside the PAGB, as shown in Figure 7.

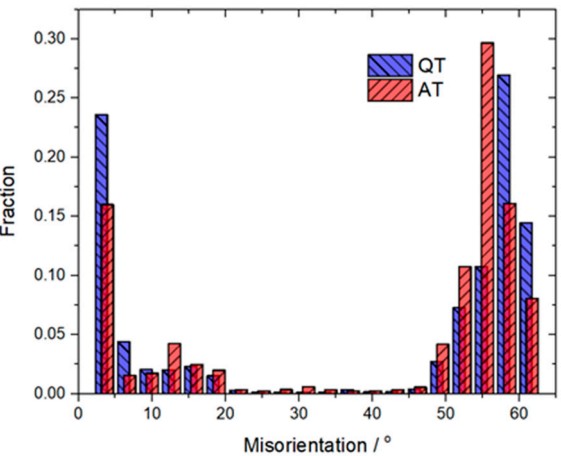

**Figure 11.** Misorientation distribution between the QT and AT specimen.

In Figure 11, the QT specimen has a higher misorientation distribution fraction than the AT specimen at an angle of less than 10°, and at 58° to 61°; an opposite tendency was observed at 46° to 55°. Consequently, the high fraction of the low-angle boundary in the QT specimen is associated with the martensite formation mechanism.

Compared to bainite, when martensite is generated, the shear stress field is larger and cannot grow easily in the same direction. Consequently, the crystallographic direction undergoes a minute change to eliminate the local stress field, after which it grows in the deformed direction at the time of growth.

This phenomenon is repeated, and two lath variants appear together [11–15,19]. Therefore, the short laths of the adjacent orientation of the QT specimen are presented in Figures 5 and 6. As a result, many low-angle boundaries appear between adjacent laths.

By contrast, bainite has a relatively low influence on the stress field compared to martensite and has fewer nucleation sites to maintain isothermal conditions during growth. Consequently, bainite produces fewer nuclei than martensite, and bainite lath grows with stronger linearity than martensite. At this time, the orientation between bainite laths is very different, and thus they have numerous high-angle boundaries rather than low-angle boundaries. This result was in good agreement with the results of other studies [11,15].

In general, grain boundaries interfere with the movement of dislocations. However, if the boundary is less than 15°, the ability to block the movement of dislocations will be reduced, and it will not act as a strengthening mechanism. However, it has been reported that the ability to block the movement of the dislocation is excellent in the case of a high-angle boundary over 15°.

Therefore, the AT specimen of the bainite lath with a high-angle boundary, as shown in Figure 11, exhibits excellent elongation and toughness compared to those of the QT specimen. Moreover, in the case of the AT specimen, because many high-carbon-retained austenites exist, it is considered to help improve the high tensile strength, excellent elongation, and fracture toughness.

### 3.2.4. Quantitative Analysis of Length and Width of the Laths

Table 1 compares the length and width of the laths analyzed using the IPF maps in Figure 5 (QT specimen) and Figure 7 (AT specimen). Here, the lath length of the AT specimen was measured with a maximum of 29.4 μm, a minimum of 7.1 μm, and an average of 15.5 μm, and the width of the AT lath was measured with a maximum of 5.29 μm, a minimum of 0.59 μm, and an average of 1.76 μm. In addition, the lath length of

the QT specimen was measured with a maximum of 9.37 μm, a minimum of 2.2 μm, and an average of 3.06 μm, and the width of the QT lath was measured with a maximum of 2.76 μm, a minimum of 0.55 μm, and an average of 0.81 μm. Here, the length of the lath of the AT specimen was approximately five times that of the QT specimen, and the width was approximately twice that of the QT specimen.

**Table 1.** Comparison of the average length and width of the laths between QT and AT specimens.

| Specimen | QT | | AT | |
|---|---|---|---|---|
| | Length (μm) | Width (μm) | Length (μm) | Width (μm) |
| Average | 3.06 | 0.81 | 15.5 | 1.76 |
| Max. | 9.37 | 2.76 | 29.4 | 5.29 |
| Min. | 2.20 | 0.55 | 7.1 | 0.59 |

The average length and width of the lath of the AT specimen were longer and wider than those of the QT specimen. The good lath properties of bainite AT specimens were in good agreement with the tendency that mechanical properties such as tensile strength, hardness, elongation, and impact energy are better than those of the martensite QT specimen, and it is judged as a good parameter to predict these properties.

*3.3. Tensile Test*

The stress–strain curves of the tensile test are shown in Figure 12, and the test results are summarized in Table 2. The stress–strain diagrams of the two AR specimens are almost the same, as shown in Figure 12a, with a maximum tensile strength of 991.5 MPa, yield strength of 526.5 MPa, and an elongation of 19.5%. Tensile strengths of QT and AT specimens are measured at 1723.1 MPa and 1824.4 MPa, 73.8% and 84% higher than the AR specimens.

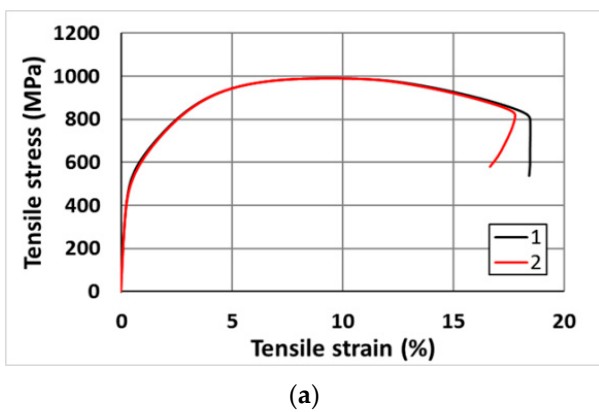 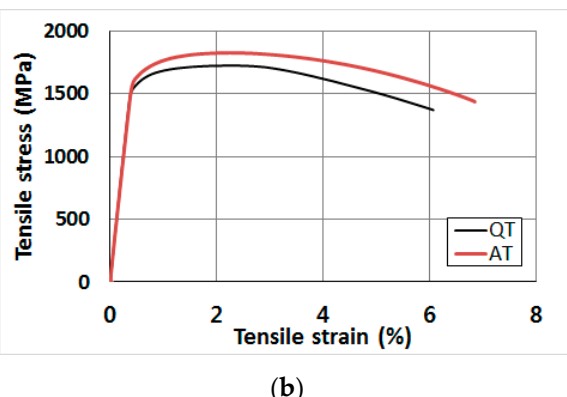

(**a**)          (**b**)

**Figure 12.** Stress–strain curves of the spring steel. (**a**) AR and (**b**) QT and AT specimens.

**Table 2.** Mechanical properties.

| Specimen | Tensile Strength (MPa) | Yield Strength (0.2%, MPa) | Elongation (%) | Reduction in Area (%) |
|---|---|---|---|---|
| AR | 991.5 | 526.5 | 19.5 | 55.3 |
| QT | 1723.1 | 1612.1 | 12.1 | 39.7 |
| AT | 1824.4 | 1681.5 | 13.4 | 49.7 |

In addition, the bainite AT specimen showed a 5.9% higher tensile strength than the martensite QT specimen. The increase in tensile strength of AT and QT compared to the

AR specimen resulted from the change in microstructure owing to the formation of bainite and martensite by heat treatment (Figure 1).

The elongation and area reduction predominate despite the increase in tensile strength. However, this property was found to be more enhanced in the bainite AT specimen than in the martensite QT specimen. The characteristics of this physical property depend on the size and shape of the laths.

### 3.4. Hardness Test

Table 3 summarizes the average Vickers hardness values of the specimens used in this study. The AR specimen showed a standard deviation (S.D) of 12.7, the QT specimen showed 11.4 and 12.5, and the AT specimen showed the lowest values of 10.1 and 10.5. The martensite QT and bainite AT specimens showed an increase in hardness of 84.5% and 102.6%, respectively, compared to the AR specimen.

**Table 3.** Results of Vickers hardness test.

| Specimen | AR | QT | | AT | |
|---|---|---|---|---|---|
| No. | #1–#3 | #1 | #2 | #1 | #9 |
| Hv | 276.1 | 511.2 | 507.6 | 559.6 | 559.1 |
| Average | 276.1 | 509.4 | | 559.4 | |
| S.D. | 12.7 | 11.4 | 12.5 | 10.1 | 10.5 |

In addition, the AT specimen showed a 9.8% increase compared to the QT specimen. The significant increase in AT and QT hardness values compared to that of the AR specimen resulted from the mixed microstructure of bainite and martensite and the microscopic formation of the structure.

### 3.5. Impact Test

#### 3.5.1. Variation of Impact Fracture Behavior according to Microstructure and Notch Shape

Figure 13a compares the Charpy impact energy (J) of the KS standard V-type notched specimen of untreated AR spring steel based on impact time (ms). The maximum impact value of these five AR specimens was 7.451 J (peak time 0.291 ms).

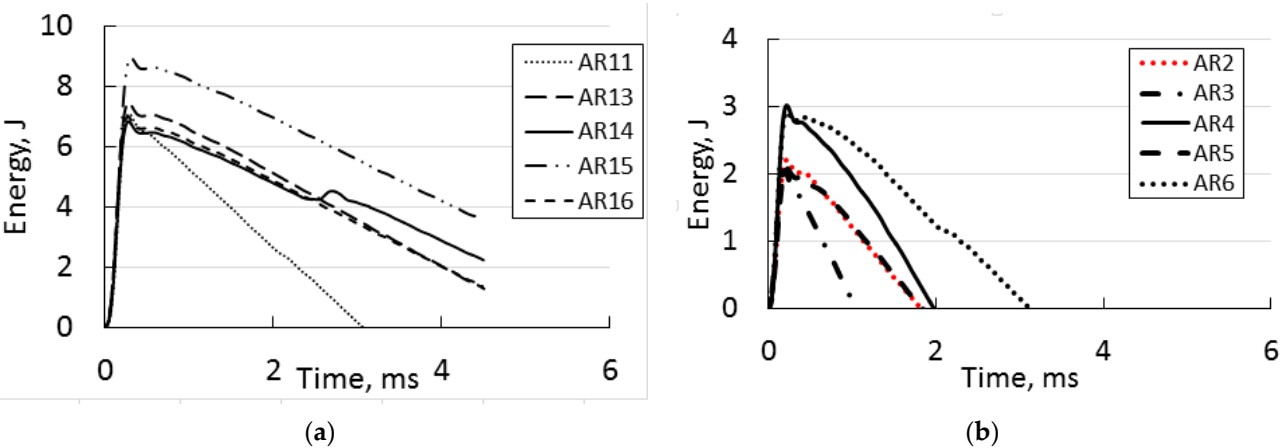

(**a**)  (**b**)

**Figure 13.** Comparison of Charpy impact toughness of the untreated AR spring steel of (**a**) standard V-type notch, and (**b**) sharp I-type notch.

Figure 13b compares the maximum Charpy impact energy of the AR specimen with an I-type notched specimen that has twice the sharp notch of a standard V-type notched impact specimen as a function of impact time. The maximum impact energy of the AR specimen is 2.436 J (peak time 0.218 ms). This was 67.3% smaller than that of the standard

V-type notch. From the above results, it is expected that the untreated AR spring steel will have lower impact toughness when fatigue cracking occurs.

Figure 14a compares the relationship between the maximum Charpy impact energy and impact time for QT martensite spring steels based on the notch type. The average maximum impact energy of standard V-type notched QT specimens (QT11, QT12) is 9.51 J, and the average maximum impact energy of sharp I-type notched QT specimens (QT1, QT2) was 7.59 J. Therefore, the maximum impact toughness of I-type notched QT specimens was reduced by approximately 20.2% compared to that of the standard QT impact specimen.

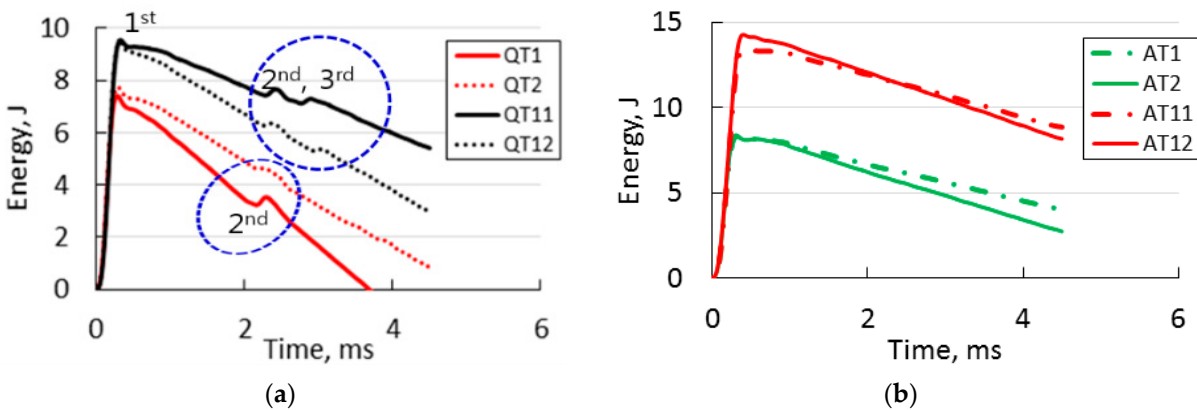

(a)　　　　　　　　　　　　　　　(b)

**Figure 14.** Comparison of Charpy impact toughness between (**a**) QT and (**b**) AT specimens with the standard V-type notch and sharp I-type notch.

Figure 14b shows the relationship between the maximum Charpy impact energy and impact time for the austempered bainite AT specimen based on the notch type. The standard V-type notched AT specimens (AT11, AT12) have an average maximum impact energy of 13.82 J, and the I-type notched AT specimens (AT1, AT2) have an average maximum impact energy of 8.35 J. Therefore, the maximum impact toughness of I-type notched AT specimens is reduced by approximately 39.6% compared to the standard AT specimens.

As shown in the figure, the impact toughness of the notch shape on the dual-sensitivity I-type impact specimens of the QT and AT specimens reduces the impact toughness by 20.2% and 39.6%, respectively, compared to the standard V-type notch specimens.

QT specimens with martensitic structures had lower impact energy than bainite AT specimens. However, the second (sharp I-type notch specimen) and third (standard V-type notch specimen) curve deformations occurred on the impact energy and time curves of the QT specimen Figure 14a). This phenomenon results from the impact of brittle cracking (Figure 17) that occurs on the surface and inside, and shows a slight fluctuation in impact energy. However, it is not possible to determine any macroscopic difference in the shock wavefront between the secondary and tertiary waves.

However, in the case of the AT specimen (Figure 14b), the impact energy decreased smoothly after the first impact energy deformation. However, one (AR14) of the AR specimens showed a secondary fluctuation (Figure 13a), and the other AR specimen had a smooth decrease in impact energy after the first impact energy fluctuation. Consequently, in the AT and AR specimens, the shear lips and impact brittle cracking (Figure 17) that occurred in all QT specimens were not observed at the fracture surface.

### 3.5.2. Variation in Charpy Impact Toughness Based on a Microstructural Change

### (1)　Case of KS Standard Specimen

Figure 15a compares the Charpy impact energy of the KS standard specimen (V-type notch) in AR, QT, and AT spring steels as a function of impact time. The average maximum impact values are 7.45 J, 9.51 J, and 13.82 J for AR, QT, and AT specimens, respectively.

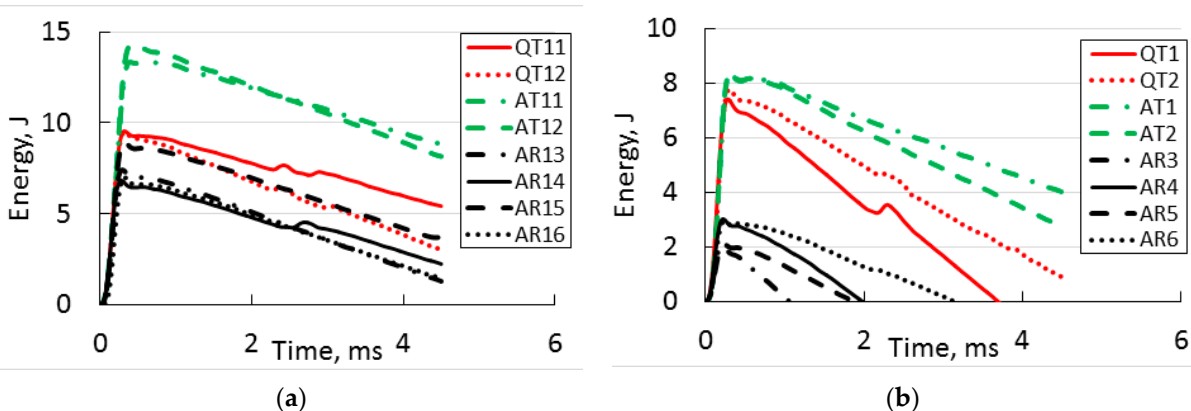

**Figure 15.** Comparison of Charpy impact toughness for AR, QT, and AT specimens. (**a**) standard V-type notch and (**b**) sharp I-type notch.

Therefore, the maximum impact energy of the V-type notch specimen increased in the order of AR < QT < AT. Moreover, the bainite AT specimen had an 85.5% larger value than the untreated AR specimen, the martensite QT specimen had a 27.6% larger value than the AR specimen, and the AT specimen had a 45.3% larger value than the QT specimen. As described above, the AT material formed from the bainite structure by heat treatment has the largest impact energy among the three specimens. Therefore, the bainite microstructure is more industrially effective than the martensite microstructure.

(2)    Case of Sharp I-type Notch Specimen

Figure 15b compares the maximum Charpy impact energy curves of AR, QT, and AT specimens with sharp I-type notches depending on the impact time. The average maximum impact toughness values for AR, QT, and AT specimens are 2.44 J, 7.59 J, and 8.35 J, respectively.

Therefore, the maximum impact energy increased in the order of AR < QT < AT specimens. That is, the bainite-structured AT specimen was 242.8% larger than that of the untreated AR specimen, and the martensite QT specimen was 211.6% larger than that of the AR specimen. Accordingly, the bainite AT specimen with a sharp I-type notch showed the highest impact energy, approximately 10% higher than that of the QT specimen.

Analysis of the notch shape in Figure 15 shows that a blunt notch requires more impact energy. In the case of fatigue cracks, the crack tip is more sensitive than a sharp I-type notch; thus, it is possible to predict that failure may occur even at low impact energies. However, the bainite AT specimen with long and wide laths was found to have a 45.3% increase in impact energy over the martensite QT specimen with short and narrow laths. Through this study, it was confirmed that the lath type is closely related to impact energy.

3.5.3. Comparison between the Charpy Impact Energy and Tensile Strength

Figure 16a compares the maximum Charpy impact energy and tensile strength according to the microstructure and notch shape of the impact specimens. That is, the maximum impact energy of the KS standard V-type notched specimens and that of the sharp I-type notched specimens of AR, QT, and AT spring steel increased almost linearly with the tensile strength.

These data show that the maximum Charpy impact energy of heat-treated QT and AT specimens increases significantly with tensile strength compared to the untreated AR specimen. They also show that the J value of the sharp I-notch specimens is significantly lower than that of the standard specimens.

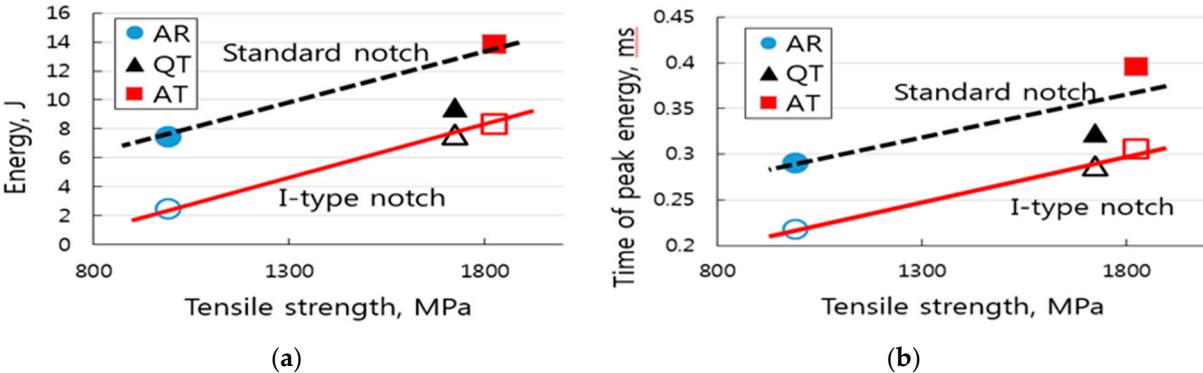

**Figure 16.** (**a**) Relationship between the tensile strength and the maximum Charpy impact energy and (**b**) relationship between the maximum impact time (ms) and tensile strength.

Vatavuk et al. compared and studied the impact fracture behavior by heat treatments at the same surface hardness level using high-carbon steel because hardness influences the impact toughness [20]. However, in this study, it is assumed that the tempered samples with the bainite microstructure obtained at a constant temperature showed lower impact properties than the QT samples.

Figure 16b compares the maximum impact time (ms) and tensile strength during an impact test, depending on the microstructure and notch shape. That is, the maximum impact time of KS standard V-type notches and sharp I-type notch specimens of AR, QT, and AT spring steels increased almost linearly with the tensile strength.

The impact toughness and peak time during the impact test in Figure 16a,b increase in the order of AR < QT < AT for both the standard V-type notch specimens and the sharp I-type notch specimens. In addition, peak times increased in proportion to the tensile strength, such as the maximum impact energy. The linearity between the tensile strength of the J value and the sensitivity of the notch shape is considered an important property value in terms of engineering and industrial applications of spring steel. In particular, the sharp I-type notch data are more linear than the standard V-type notch data because the sensitive notch produced less scattering in the experimental results.

3.5.4. Fracture Analysis of Impact Specimens

(1)　Martensite QT Specimen

Figure 17a shows a low-magnification (30×) SEM photograph of a QT specimen (standard V-type notch) after impact testing, which was assembled by taking six pieces for macroscopic inspection of the entire fracture surface. In this photograph, cleavage surfaces are mainly observed on the surface of the fracture. The unique fracture surface of the four martensitic QT specimens used in this study had no significant deformation of the specimen owing to ductility. In addition, the fracture surface shows a grain boundary brittleness fracture without dimples, and shear lips (indicated by three white arrows) are partially formed on the three surfaces.

As shown in Figure 17a, an internal brittle crack (indicated by a red arrow) with a depth of approximately 5.39 mm was formed by embrittlement and cracking (5,6). In addition, this internal impact brittle crack was a semi-elliptical crack formed up to 55 mm in length (corresponding to Figure 17d ①②) during the impact test. That is, Figure 17a,b show that an intruding shear lip shape was formed, and an extruded shear lip shape was formed on the other side. In this study, the shear lip shapes of cracks inside this specimen are referred to as intrude shear lip (ISL) and extrude shear lip (ESL).

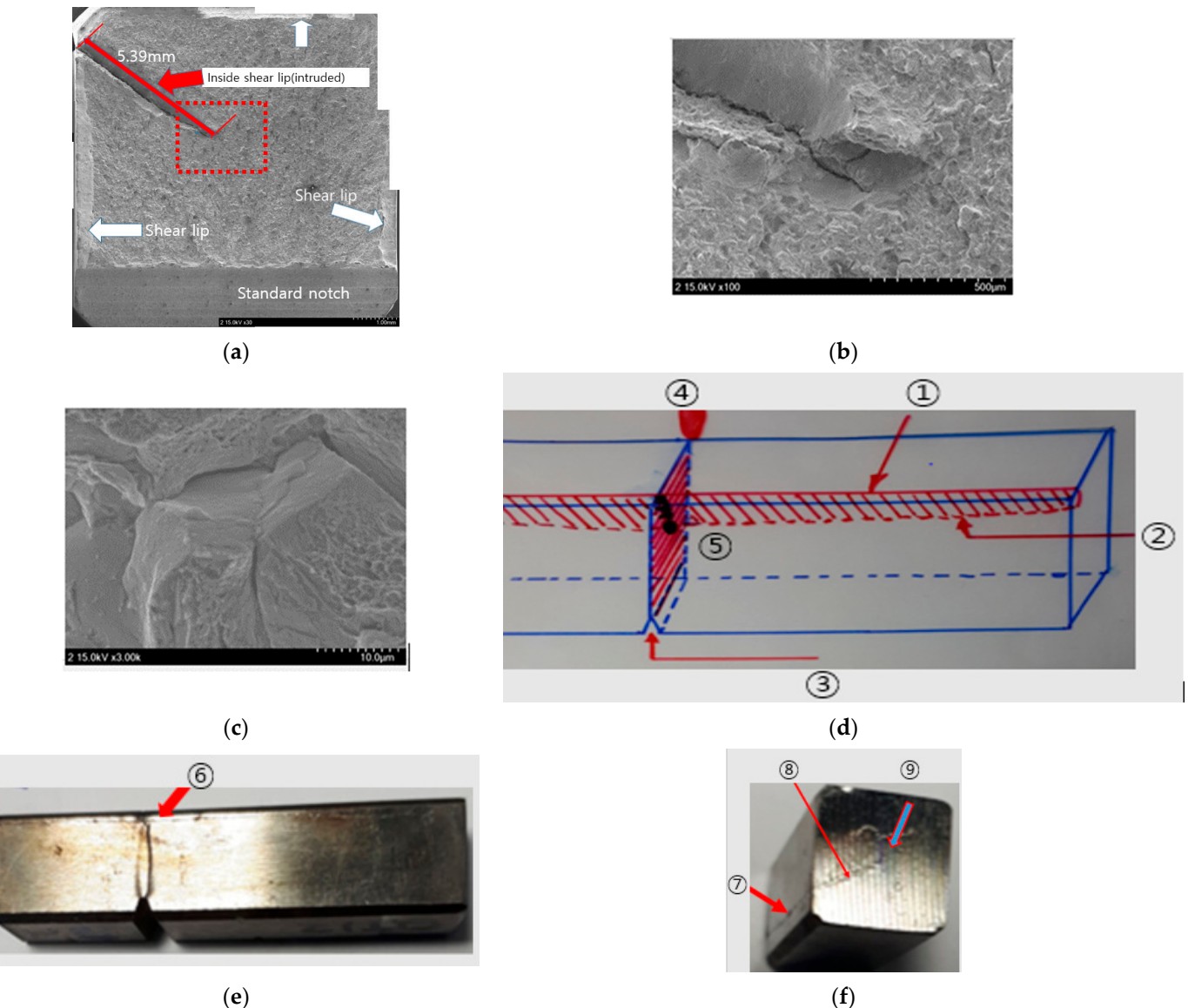

**Figure 17.** Charpy impact fractured surfaces of the QT specimen. (**a**) 30×, (**b**) 100× and (**c**) 3000×. (**d**). A schematic of the impacted specimen of (**e**), ① indicates a semi-elliptical brittle surface crack formed up to 55 mm in length and 5.39 mm in depth during the impact test. The part indicated by the diagonal line ② is the internal shape of the semi-elliptical brittle surface crack, ③ is a notch and is a fracture surface made by the impact test, ④ indicates a knife edge of the impact tester, ⑤ shows the same fracture surface as (**a**), indicating that a brittle crack with a depth of approximately 5.39 mm propagates instantaneously in the overall length (55 mm) direction of the specimen. (**e,f**) Arrows (⑥⑦) indicate a long outside brittle surface crack formed by the impact test (corresponds to (**d**) (①)) ⑧ indicates the depth of the crack of approximately 5 mm (corresponds to (**d**) (②)). Arrow ⑨ in photo (**f**) indicate the crack tip.

The area indicated by the dotted rectangle in Figure 17a enlarges the crack tip, as shown in Figure 17b,c. The rectangle in Figure 17a is magnified to 100× in Figure 17b, and the tip of the ISL-type crack can be observed. The center of Figure 17b is magnified from Figure 17c to 3000× to show the grain boundary brittleness fracture and the tear-type crack front surface.

Figure 17d shows a schematic of the impacted specimen (Figure 17e) to describe the shape of the surface and internal brittle cracking caused by the impacted martensite QT specimen. When an impact is applied by knife-edge ④, the impact specimen is instanta-

neously broken and separated at the notched portion ③. The separated fractured surface of the hatched rectangle ⑤ is shown in Figure 17d. Owing to hydrogen embrittlement, a semi-elliptical brittle crack was generated over the entire length of the impact specimen (corresponding to Figure 17d ①②) at a depth of approximately 5.4 mm. The internal shape of the surface crack is displayed obliquely, as indicated by arrow ②.

Brittle cracking owing to this impact is also visible to the naked eye on the surface ⑥⑦, and this surface crack is marked with arrows (Figure 17e). Next, the length and shape of the observed internal crack are shown on the OM (Figure 17f) to confirm that the crack tip of the internal crack cut from the fracture specimen was longer than 5 mm. From Figure 17d–f, the presence of brittle cracks on the surface owing to the impact energy of the martensitic QT specimen was confirmed.

In all QT specimens, such as those from Figure 17a–f, ESL and ISL with a depth of approximately 5 mm were formed from the inside. That is, in the ISL type, a long brittle surface crack with a length of 55 mm was instantly formed by the impact energy. The formation of these shear lip-type internal cracks resulted in embrittlement and cracking [5,6]. The systematic initiation mechanism of these internal cracks is believed to be necessary for further studies.

(2)    Bainite AT specimens

Figure 18a shows a connection of nine SEM photographs (30×) for macro observation of the impact fracture surface of the bainite AT specimen. The width of Figure 18a is the width of the specimen 10 mm. No internal brittle crack, as shown in Figure 17a, was observed in the impact fracture surface of all AT specimens. In addition, in the martensite QT specimen, many shear lips were generated from the three surfaces; however, in the bainite AT specimen, a small local generation was observed, and the ductility was high.

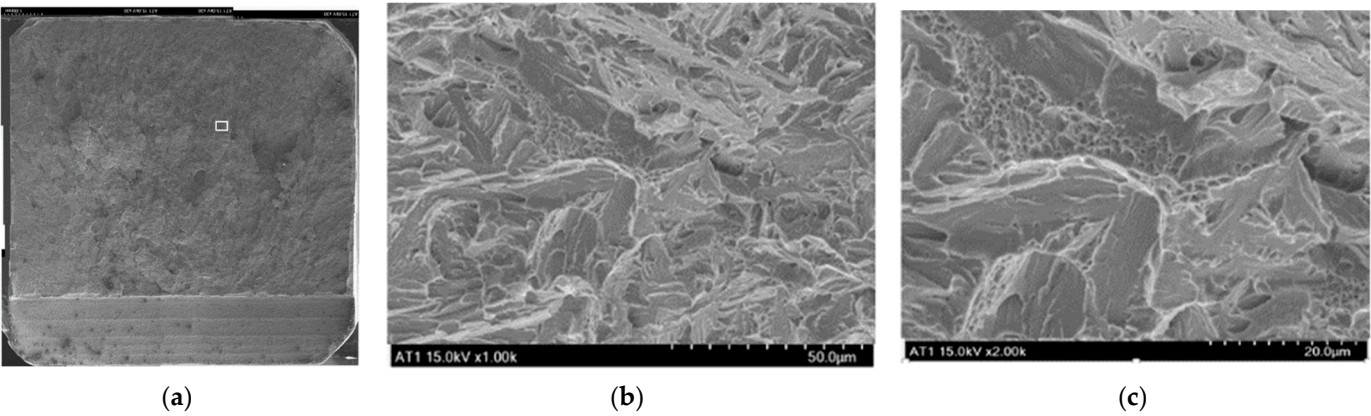

(a)           (b)           (c)

**Figure 18.** Charpy impact fractured surface of AT specimen. (**a**) indicates that shear lip is small (30×), (**b**) 1000×, and (**c**) 2000× is enlarged SEM of the area indicated by the rectangle near the center of Figure 18a.

Figure 18b (1000×) and c (2000×) are enlarged SEM photographs of the area indicated by the rectangle near the center of Figure 18a. Here, many laths observed in Figure 7a were also observed, and many fine ductile dimples were observed between the laths. In addition, there is almost no dimensional change or plastic deformation of the specimen owing to impact.

## 4. Conclusions

In this study, we analyzed the process of transforming bainite to improve its spring life and quality, and systematically studied the impact characteristics and mechanical property changes according to the length of the lath and the width of the microstructure. The following results were obtained:

(1)　EBSD quantitative analysis revealed that QT exhibited a heterogeneous lath size and cementite distribution; however, the AT had a homogeneous lath size and cementite distribution. The QT martensite structure had a high fraction of the low-angle boundary, whereas the AT bainite structure had a high fraction of the high-angle boundary and had many high-carbon retained austenites, exhibiting excellent elongation and toughness compared to those of the martensite QT.

(2)　EBSD and SEM confirmed that AT transformed into a bainite structure after the austempering cycle. Quantitative analysis of laths of bainite AT showed that their dimensions were approximately five times longer and two times wider than those of QT with a martensitic structure. AT with long and wide laths had improved mechanical properties such as tensile strength, hardness, impact toughness, and elongation rate compared to QT.

(3)　The tensile strengths of QT and AT were 1723.1 MPa and 1824.4 MPa, which were 73.8% and 84% higher than untreated AR (991.5 MPa), respectively. In addition, the bainite AT exhibited a 5.9% higher tensile strength than the martensite QT. The hardness of QT and AT increased by 84.5% and 102.6%, respectively, compared to that of AR. In addition, the hardness of AT increased by 9.8% compared to that of martensite QT.

(4)　The maximum impact energy of the KS standard V-type notch specimen increased in the order of AR < QT < AT, the AT specimen was 85.5% larger than AR, and QT was 27.6% larger than AR. In particular, the bainite AT made of long and wide laths had a 45.3% higher impact energy than the martensite QT made of short and narrow laths. The AT with the V-type notch specimen formed by the bainite structure had the largest impact energy among the three types of specimen.

(5)　The maximum impact energy of the sharp I-type notch specimen increased in the order AR < QT < AT. The bainite AT was 242.8% larger than the AR, and the martensite QT was 211.6% larger than the AR. Therefore, bainite AT with a sharp I-type notch had the highest impact energy of approximately 10% higher than QT. The maximum impact toughness of the I-type specimen was reduced by approximately 39.6% compared to that of the standard specimen. The impact of the notch shape on the I-type impact specimens of QT and AT reduced the impact toughness by 20.2% and 39.6%, respectively, compared to the standard V-type notch specimen.

**Author Contributions:** Data curation, M.-S.S.; Formal analysis, N.-K.P.; Funding acquisition, M.-S.S. and S.-H.N.; Investigation, C.-M.S. and N.-K.P.; Methodology, C.-M.S. and N.-K.P.; Project administration, S.-H.N.; Writing—original draft, C.-M.S. and N.-K.P.; Writing—review & editing, C.-M.S. and N.-K.P. All authors have read and agreed to the published version of the manuscript.

**Funding:** This research was supported by the Development of Reliability Technology of Standard Measurement for Hydrogen Convergence Station funded by the Korea Research Institute of Standards and Science (KRISS-2021-GP2021-0007). This work was supported by the National Research Foundation of Korea (NRF 2017033524) and the National Research Council of Science & Technology (NST CAP20032-200) grant funded by the Korean government (MSIT).

**Conflicts of Interest:** The authors declare no conflict of interest.

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
