# Peer review of "Impact Toughness of Spring Steel after Bainite and Martensite Transformation"

_metals, doi:10.3390/met12020304_

Round 1

Reviewer 1 Report

The present work investigates the microstructure and mechanical properties in a spring steel, showing so many details caused by bainite and martensite transformation. The experimental work is abundant and in-depth. The results indicated that bainitic microstructure had better strength/ductility, hardness, and impact toughness, which is mainly due to sizes effect. That is valuable for designing spring steels with good performance. Some minor suggestions are as follow:

  1. In Figure 3a, please sign which is ferrite or pearlite, and which is ferrite in Figure 3b.
  2. Line 154, the author said that 23% Fe3C was formed in the AR specimen. Is the green part in Figure 4 termed as Fe3C? Notes should be given in the caption. Explanation about the morphology and amount of Fe3C should be given
  3. In Figure 9, where is cementite? Line 274-275, the author said a considerable amount of carbon forms cementite. I think a TEM micrography is better than the EBSD, to show the inside precipitation.

Author Response

In Figure 3a, please sign which is ferrite or pearlite, and which is ferrite in Figure 3b.
-->Supplement the contents of ferrite and pearlite in the explanation of Figure 3a.

Line 154, the author said that 23% Fe3C was formed in the AR specimen. Is the green part in Figure 4 termed as Fe3C? Notes should be given in the caption. Explanation about the morphology and amount of Fe3C should be given
-->Supplemented during the description in Figure 4.

In Figure 9, where is cementite? Line 274-275, the author said a considerable amount of carbon forms cementite. I think a TEM micrography is better than the EBSD, to show the inside precipitation.
-->However, cementite does not appear in the retained austenite, but it is dissolved in austenite and serves to thermodynamically stabilize it.

Reviewer 2 Report

This is an amount of effort combining lots of experiments to study strength and toughness of three heat treatment processes. The work can be published in this journal; however, a couple of major revisions is needed before accepting:

The performance index required by spring steel should be reflected in the paper, including tensile strength, impact ete. Whether the mechanical properties in current work can meet the required properties of spring steel.

Even the spring heat treatment production process parameters should be mentioned, which can be used as a basis for choosing AT, QT and AR. Did these three processes be referenced for spring steel production? Briefly, what optimization or reference can the current work provide for the production of springs.

Line 95: Fig.1: This figure shows the selected proceses, which should be additionally  described in the figure, including key phase temperature, Ms, AC1 and AC3 .

Line 157: Fig.4: the difference between two phases needs to be shown.

English needs to be improved for this manuscript. There are a number of grammatical errors and instances of badly worded/constructed sentences.

Author Response

This is an amount of effort combining lots of experiments to study strength and toughness of three heat treatment processes. The work can be published in this journal; however, a couple of major revisions is needed before accepting:
 -->Although it is a good suggestion, the purpose of this study is EBSD analyzes what type of bainite can be obtained under the heat treatment condition of Fig. 1b, and quantitatively analyzes the microstructure change. In other words, I think it is more beneficial to compare the characteristics of these bainite and martensite microstructures with each other as in this study.

Even the spring heat treatment production process parameters should be mentioned, which can be used as a basis for choosing AT, QT and AR. Did these three processes be referenced for spring steel production? Briefly, what optimization or reference can the current work provide for the production of springs.
-->This study results are one of the many conditions for producers of spring steel to produce bainite structure rather than martensite (brittleness is a problem). In other words, the producer is conducting research while analyzing experimental data for economic efficiency and effectiveness based on the research results obtained under multiple conditions. For reference, comparative data such as fatigue behaviors and wear characteristics in bainite and martensite under the same heat treatment conditions as in this paper will be published in another paper.  

Line 95: Fig.1: This figure shows the selected proceses, which should be additionally  described in the figure, including key phase temperature, Ms, AC1 and AC3 . 
--> It is not a TTT curve, so marking the desired symbol causes confusion. Please understand.

Line 157: Fig.4: the difference between two phases needs to be shown.
--> The difference between the two phases is widely known as a result of the EBSD analysis function to observe the microstructure.  That is, it is divided into image quality map, inverse pole figure map, phase map, etc.  In addition to OM, SEM, and TEM, EBSD has recently been developed, which is a very convenient era for quantitatively analyzing crystal structures in various ways.

Reviewer 3 Report

The knowledge published in this work has been known for some time. There is nothing new in this work: steel with carbide-free bainite is tougher than steel with tempered martensite and ferrite-pearlite. One piece of new information from this work is the use of an instrumentation sensor during the Charpy test.

Author Response

The knowledge published in this work has been known for some time. There is nothing new in this work: steel with carbide-free bainite is tougher than steel with tempered martensite and ferrite-pearlite. One piece of new information from this work is the use of an instrumentation sensor during the Charpy test.

-->Thanks for your comments

Reviewer 4 Report

The manuscript comprises research work on spring steel. The main goal has been to examine how different heat treatments and subsequent microstructures affect the impact toughness and other mechanical properties of the steel. Part of the work has been conducted well, for example the EBSD characterization and microstructural analysis. However, there are too many problems with the scientific output of the paper.

First of all, the headline states “impact behaviors”. This is incorrect English and the term “impact” may refer to many things, such as impact wear or other type of testing, which includes impacts. Toughness or impact toughness would be more suitable.

Secondly, there are problems with English grammar and problems with the references (both use and numbering). Some vital information is missing (number of samples, quenching rate, testing temperature etc.) from the experimental section and some of the metallurgical terms used are incorrect (“old austenite”). Too many of these types of errors are standing out in the text.

However, the main problem is that quite bold conclusions have been drawn from such a small number of specimens. Not only that, but also the comparison between AR and the other too variants is made too dramatic with the use of percentage difference: in absolute numbers, the differences are minor in impact toughness. Metallurgically, there are also some major problems: the tempering temperature for QT variant is quite bizarre, why 430 °C? This could be in the tempered martensite embrittlement range for this specific steel. More tempering treatments should have been done to discover the best suitable option, not only some manufacturer recommendation. The same applies for the AT steel: was this the best option? Have the other processing routes been tested? In addition, the conclusions regarding lath size and impact toughness are quite contradictory to the literature.

As a summary, the comparison between these steels is obsolete, both metallurgically and statistically. Therefore, much more work is needed for consideration for publication of the current work. The paper cannot be recommended for publication in the current state and should be rejected.

Author Response

Thanks for your comments. 

  1. First of all, the headline states “impact behaviors”. This is incorrect English and the term “impact” may refer to many things, such as impact wear or other type of testing, which includes impacts. Toughness or impact toughness would be more suitable.
    Instead of “impact behaviors”, “impact toughness” seems like a good suggestion.
  2. Secondly, there are problems with English grammar and problems with the references (both use and numbering). Some vital information is missing (number of samples, quenching rate, testing temperature etc.) from the experimental section and some of the metallurgical terms used are incorrect (“old austenite”). Too many of these types of errors are standing out in the text
    English is not important, but the contents of the research material are important. English is not the author's native language.
  3. However, the main problem is that quite bold conclusions have been drawn from such a small number of specimens. Not only that, but also the comparison between AR and the other too variants is made too dramatic with the use of percentage difference: in absolute numbers, the differences are minor in impact toughness. Metallurgically, there are also some major problems: the tempering temperature for QT variant is quite bizarre, why 430 °C?
    This study results are one of the many conditions for producers of spring steel to produce bainite structure rather than martensite (brittleness is a problem). In other words, the producer is conducting research while analyzing experimental data for economic efficiency and effectiveness based on the research results obtained under multiple conditions.
  4. This could be in the tempered martensite embrittlement range for this specific steel. More tempering treatments should have been done to discover the best suitable option, not only some manufacturer recommendation. The same applies for the AT steel: was this the best option? Have the other processing routes been tested? In addition, the conclusions regarding lath size and impact toughness are quite contradictory to the literature.
    The purpose of this study is to obtain “quantitative data” that researchers usually miss and to utilize it for engineering purposes. So, I took the path that nobody else did. That is, the relationship between lath size, shape and impact toughness.

Round 2

Reviewer 2 Report

A large number of experiments carried out and corresponding results were obtained.  There is a suggestion that should be considered is that the experimental design should take full account of actual production.  Such  studies would make more sense.

Author Response

Complement in the text

Reviewer 4 Report

The authors have replied the reviewer's comments, but the response is not satisfactory. The authors claim to have "quantitative data", but this is contradictory to the paper: only few specimen have been tested and characterised. This would be appropriate with some new and novel materials, but the current research work does not represent such research. 

Though the instrumented data from Charpy-V is indeed novel, there is generally not enough novelty in the paper for publication in the current form. More tests with different steel conditions should be conducted in order to confirm the findings, especially for the "quantitive data". The reported findings do not suffice for a scientific article.

Author Response

Obtaining the "quantitative data" proposed by the reviewer is a very good opinion and very sympathetic to myself, and I would love to get it too. However, one study requires a lot of time, financial support, human support, etc., but it is limited. In future research, I will plan and work hard to obtain many "quantitative data" by carefully referring to the opinions of reviewer.